# Healthcare Providers’ Perceptions about the Role of Spiritual Care and Chaplaincy Services in Substance Use Outpatient Treatment

**DOI:** 10.3390/ijerph19159441

**Published:** 2022-08-01

**Authors:** Brian S. W. Earl, Anne Klee, Ellen L. Edens, James D. Cooke, Holly Heikkila, Lauretta E. Grau

**Affiliations:** 1Departments of Chaplain Services and Mental Health Service Line, Veterans Administration Hospital, West Haven, CT 06516, USA; brian.earl@va.gov (B.S.W.E.); anne.klee@va.gov (A.K.); ellen.edens@va.gov (E.L.E.); james.cooke2@va.gov (J.D.C.); holly.heikkila@va.gov (H.H.); 2Chaplain Services, Veterans Administration Hospital, Albuquerque, NM 87108, USA; 3Department of Psychiatry, Yale School of Medicine, New Haven, CT 06511, USA; 4Chaplain Services, Milwaukee Veterans Administration Hospital, Milwaukee, WI 53215, USA; 5Spiritual Life Office, University of Chicago, Chicago, IL 60637, USA; 6Department of Epidemiology of Microbial Diseases, Yale School of Public Health, New Haven, CT 06511, USA

**Keywords:** chaplaincy, religion, spirituality, spiritual care, outpatient substance use treatment, veteran services

## Abstract

Addressing patients’ religion and spirituality (R/S) needs has been associated with positive health outcomes. However, despite receiving extensive training in spiritual assessment and care, chaplaincy services are primarily confined to inpatient settings, with few studies occurring in outpatient settings. The study sought to understand mental health providers’ views about what shaped provider and patient motivation to engage in R/S discussions and seek referrals to chaplaincy services. We conducted five one-hour focus group sessions with a total of 38 staff members and thematically analyzed the resulting session and field notes. We identified four themes concerning provider knowledge and attitudes about R/S and chaplaincy services: Staff Information Needs, Staff Motivation to Discuss R/S and Refer, Patient Motivation to Use Chaplaincy Services, and Chaplain Accessibility. The study findings suggest that providers in outpatient substance use treatment clinics in the Veterans Health Administration are receptive to learning about R/S care and the possibility of expanding chaplaincy services. However, staff have misconceptions about the roles and responsibilities of chaplains. Attitudes about and experiences with R/S discussions varied. Trust and confidence in the benefits of chaplaincy services may be improved among both providers and patients by increasing chaplains’ accessibility and visibility within these outpatient settings.

## 1. Introduction

Healthcare chaplains are trained to work as members of interdisciplinary healthcare teams to assist patients, families, and staff with their religious and spiritual (R/S) needs [1,2]. The integration of chaplains into healthcare is positively associated with patient outcomes and satisfaction, employee engagement and retention, and savings in healthcare spending, regardless of whether patients directly request chaplaincy services [3,4,5,6,7]. Chaplains working in medical centers have a graduate theological degree, faith group ordination, and 1600 h of Clinical Pastoral Education in a healthcare setting [8]. In the Veterans Health Administration, chaplains provide “in-depth assessment, evaluation, and treatment of patients”, developing “close working relationships with staff members of other professional health care disciplines” [9]. However, R/S discussion and referral to chaplaincy services appear to vary greatly across providers and settings. While few patients request to see a chaplain [10], many who seek to discuss their spiritual needs prefer to meet with someone specifically trained in this area instead of their physician, as many in this latter group report being uncomfortable discussing R/S topics and appreciate the ability to refer their patients to chaplains [1,11]. Despite physicians’ discomfort, most patients express a desire to discuss R/S needs with a healthcare provider [12] and more so in cases involving serious diagnoses [13].

Studies on the activities and functions of chaplains themselves are largely limited to inpatient settings, with common exceptions being outpatient palliative care and oncology services [14,15,16,17,18,19]. Few chaplains work on outpatient mental health teams [20,21] despite the relevance of R/S in mental health [22,23,24,25,26]. Limited chaplaincy presence on mental health teams is unfortunate given that psychiatrists are more likely to encounter spirituality issues than are other physicians [27,28]. Compared to other mental health providers, psychiatric nurses were less likely to encourage R/S discussions but more likely to refer patients to chaplaincy services instead [28,29], although chaplains opined that mental health staff only occasionally discussed R/S or referred patients to chaplains [28].

Studies that examine the role of spirituality, religion, or chaplaincy services in treating substance use disorder (SUD) in outpatient specialty clinics are limited. Most involve twelve-step facilitation programs (e.g., Alcoholics Anonymous), not typically led by chaplains [2,30]. The findings from these studies suggest, however, that R/S should be considered when evaluating long-term outcomes [31,32] and that spirituality may play a more positive role in recovery than religiosity [33].

Despite the presence of a chaplain in outpatient SUD treatment clinics, chaplaincy services appeared to be underutilized. Furthermore, given the limited published information about chaplaincy service utilization in outpatient settings, the study sought to understand the reasons that may account for this underutilization. Recognizing the value of R/S in mental health and embedding chaplains in the outpatient SUD service clinics [34] and reasoning that outpatient mental health providers play a potentially critical role in linking patients to chaplaincy services, we conducted focus group discussions with these providers to understand their perceptions about R/S and chaplaincy service utilization in outpatient SUD treatment clinics and the potential barriers and facilitators to using these services.

## 2. Materials and Methods

### 2.1. Setting

This qualitative study was conducted within the VA Connecticut Healthcare System (VACHS) in the Substance Use Specialty Care outpatient clinics located in West Haven, Connecticut, USA, that provide medical and/or psychosocial services for 1200 SUD patients annually. The VACHS West Haven Division includes a 216-bed inpatient facility and ambulatory care clinics at the main campus and a nearby facility that offers community-based rehabilitative programs, including day treatment, crisis intervention, and housing programs.

### 2.2. Study Sample and Procedures

We contacted the SUD outpatient teams via email, telephone, and/or in-person conversations to schedule focus group sessions during times when most staff were available to meet (e.g., team meetings). All focus groups occurred between March and July 2019. Inclusion in the study was limited to those individuals responsible for providing direct substance use-related, non-acute care to outpatients. They included psychiatrists, psychiatry fellows, psychologists, social workers (SWs), addiction therapists, pharmacists, and peer specialists (PSs). In the clinics, psychiatrists are primarily responsible for prescribing, monitoring medications, and overall management of patients’ clinical care. Psychologists, SWs, and addiction therapists provide psychological, emotional, and social support for patients. PSs provide supportive care to patients. Although considered to be a convenience sample, of the five programs contacted, only one staff member was unable to participate in any of the scheduled focus groups. Four focus groups included participants from different disciplines, the fifth group consisted uniquely of PSs. We felt that it would be important to “oversample” this group, as these individuals are sometimes the staff with the closest and most frequent contact with the Veteran patients. In addition, many PSs themselves have histories of substance use disorders. As such, they may have unique and potentially different perspectives about spiritual care than the more traditional health care providers on the team.

Each focus group session lasted approximately 60 min. Two members of the study team assisted at each session (BE, JC, LG, AK, EE), serving as either group moderator or scribe, except for the PS session, where one member (BE) assumed both responsibilities. The sessions were not audio-recorded. Instead, the two team members met immediately after each session to debrief, describe the content covered during the session, and include any verbatim quotes from the session notes. Documents created for each post-session discussion served as the data for analysis.

The study was deemed a quality improvement study and therefore exempt from human subjects research review by the IRB at VACHS, although publication or presentation describing the study is permitted.

### 2.3. Data Collection and Analysis

The discussion guide for the study included questions on three topics: (a) staff understanding of R/S and chaplaincy services, (b) contextual factors that can contribute to provider decisions to discuss R/S and/or refer patients to chaplaincy services, and (c) recommendations to promote use of chaplaincy services. The post-session reports were independently reviewed by the entire team, who met regularly to develop the codebook; once finalized and applied to two reports, the remaining were then coded by the senior author and entered into ATLAS.ti (Version 7.1.7, ATLAS.ti Scientific Software Development GmbH, Berlin, Germany).

A total of 16 codes were generated concerning R/S discussions with patients, referral to chaplaincy services, perceived barriers and facilitators to R/S discussions or referrals, and provider and patient characteristics that may affect R/S discussions or referrals. The data were then analyzed thematically and iteratively using an inductive approach wherein patterns in the data across the five focus groups were organized into themes that were grounded in the data [35,36]. Negative instances where the data did not fit the existing themes were also identified as part of the confirmability process [37]. Participants did not review the analytic findings.

## 3. Results

### 3.1. Description of the Study Sample

A total of 38 staff members were interviewed across five focus groups. The majority of the sample was male (Table 1). Although no specific demographic data were collected, most (with the exception of the psychiatric fellows) had been in their current positions for over one year. No group differences in thematic content were noted by either participants’ gender or job title.

### 3.2. Conceptual Framework of the Identified Themes

Motivation to engage in a given behavior is considered a necessary precursor to engaging in health behaviors [38,39,40]. As part of the thematic analysis, we developed a conceptual framework that was firmly grounded in the data. We identified two pathways, four themes, and the relationship between themes that could lead to use of chaplaincy services. The provider-initiated pathway involved staff motivation to engage in R/S discussions with patients and/or to refer them to chaplaincy services; the patient-initiated pathway pertained to participants’ perceptions about patients’ personal motivation to use chaplaincy services and independent of the patients ever having had an R/S discussion with a provider. These two pathways shaped use of chaplaincy services and subsumed four themes: (1) Staff Information Needs, which, in turn, influenced (2) Staff Motivation to Discuss and Refer, (3) Patient Motivation to Use Chaplaincy Services which directly influenced whether patients would seek such services, and (4) Chaplain Accessibility which directly influenced both staff and patient motivation and indirectly influenced the outcome of using chaplaincy services (Figure 1). The patient-initiated pathway concerned providers’ beliefs about patients’ personal motivations to use chaplaincy services (independent of patients ever having had an R/S discussion with a provider). Both pathways were influenced by Chaplain Accessibility. These themes and the sub-themes subsumed within each will be described below as part of reporting the findings from the thematic analysis.

### 3.3. Staff Information Needs (Theme 1)

This first theme concerned the information needs of staff about R/S and chaplaincy services and included two sub-themes: (1) Information Deficits and Misconceptions and (2) Staff Desire for More Information.

#### 3.3.1. Information Deficits and Misconception Sub-Theme

Staff understood the role of chaplaincy services in cases involving bereavement or existential crises but had an inaccurate understanding of chaplains’ larger role, training, and ethical boundaries. For example, before learning that chaplains are ethically prohibited from proselytizing, one provider noted concern about proselytization. Others appeared unaware of the diverse religions and denominations represented within chaplaincy services. They also were unaware of chaplains’ specialized training in interfaith care and effective and appropriate communication skills.


*“There could be assumptions about the chaplain—the chaplain might not be comfortable with seeing someone. I know some LGBTQ+ folks are concerned about judgment. They are wondering, is the chaplain okay?”*
(Participant 3:22, Male, Psychiatrist/Fellow)

#### 3.3.2. Desire for More Information sub-theme

Providers appeared interested in receiving information about the nature and scope of chaplaincy services. They wished to learn more about chaplains’ role in healthcare, chaplains’ level of training, and chaplain-initiated interventions. Participants expressed a desire to receive follow-up information on provider-initiated referrals and outcomes.


*“I tell clients to speak with [the chaplain] but not sure if they follow up. I should make formal referrals.”*
(Participant 1:8, Female, Psychologist)


*“More knowledge about chaplaincy, more info about the referral process and when to refer, more information about how chaplains are trained and educated.”*
(Participant 322, Male, Psychiatrist/Fellow)

### 3.4. Staff Motivation to Discuss R/S and Refer to Chaplaincy Services (Theme 2)

The second theme on the staff-initiated pathway concerned elements that could promote the use of chaplaincy services. It included three sub-themes: (1) staff attitudes about discussing R/S with patients, (2) anticipated consequences of referring patients to chaplaincy services, and (3) case-dependent considerations when deciding whether to refer patients to chaplaincy services.

#### 3.4.1. Staff Attitudes about Discussing R/S with Patients Sub-Theme

Participants’ attitudes about discussing R/S with patients varied substantially. Some were passive and only engaged in R/S discussions when patients initiated these conversations.


*“I don’t ask. But if they bring it up, I do talk about it with them.”*
(Participant 2:10, Male, PS)

Others noted their willingness to engage in R/S discussions in specific instances, such as completing the required screening and monitoring assessment forms. R/S discussions occurred less often at follow-up visits and only as time permitted or when introduced by members of group counseling sessions. One nurse observed that staff seldom discuss politics or religion with patients and that R/S discussions can be uncomfortable and “might lead into conflict” with their patients.


*“Sometimes at intake I bring it up, but then it gets left to the side…other times it is relevant to patient care.*
*”*
(Participant 3:21, Male, Psychiatrist/Fellow)

By contrast, other participants stated that they actively and routinely ask patients about R/S issues. They appeared to believe that R/S could be a crucial issue for some patients and monitored for its potential treatment relevance over time.


*“I ask. Spirituality is seen as a power, healing and tool. It is important to peer support. I’m not afraid to mention.*
*”*
(Participant 2:15, Male, PS)


*“I always ask when I’m first meeting someone. It’s an important question. And then during treatment, I listen for when it comes up…When they talk about values or spirituality.*
*”*
(Participant 1:3, Female, SW)

Although most noted that R/S issues were often important to discuss, one participant believed that R/S discussions were usually unimportant to treatment and should be considered only as a last resort.


*“*
*For some people, especially those that you’ve treated for years, and if this is as good as it gets and that’s not very good, then a referral to a chaplain to learn to accept where we are in treatment seems appropriate…And the spiritual realm maybe comes in at the limits of what we can do.*
*”*
(Participant 1:6, Male, Psychiatrist/Fellow)

Participants’ own former or current R/S affiliations could also influence their attitudes about engaging in R/S discussions. Some appeared reluctant to discuss R/S when their own R/S views differed from those of their patients. Others with current R/S affiliations appeared inclined to engage in R/S discussions with patients, especially patients with similar R/S beliefs.


*“I don’t subscribe to a Judeo-Christian worldview—so I can’t relate, and it has caused problems in the past with rapport. That’s why I have been conditioned to be less curious about religion and spirituality.*
*”*
(Participant 3:21, Male, Psychiatrist/Fellow)


*“Well, if someone shares my own R/S belief I will ask them about it more. Otherwise, I just listen. If I am the same—Baptist—I may tell them that I am, and we will connect more.*
*”*
(Participant 2:13, Male, PS)

Some participants’ attitudes were shaped by previous experiences with R/S discussions. Those with previous positive experiences that resulted in positive clinical outcomes appeared more amenable to future R/S discussions with patients. By contrast, participants with previous negative experiences such as break-down of therapeutic rapport said they were less likely to initiate such discussions. For example, one participant stated that the patient-provider relationship deteriorated when a patient remarked that the participant held different R/S beliefs from the patient.


*“I’m not a Christian now...so it was awkward and it hurt the rapport we had.*
*”*
(Participant 3:23, Female, Psychiatrist/Fellow)

#### 3.4.2. Anticipated Consequences Sub-Theme

For some participants, motivation to discuss R/S or refer patients to chaplaincy services was based upon expectations about potential consequences rather than their prior experiences. Some participants worried that patients might be angered by such discussions or referrals. Others were concerned that R/S discussion in group settings might promote divisiveness or that the group leader may appear to show favoritism to a particular R/S viewpoint. One participant appeared to initiate R/S discussions as a means of self-protection, that is, to alert patients not to inadvertently make disparaging remarks about the participant’s religion.


*“I won’t even bring the subject up if I suspect it will anger them.*
*”*
(Participant 5:30, Female, Nurse)


*“So I ask [the patients], “Do you believe in God?” And they tell me and then of course they ask me, “Do you believe in God?” And I tell them I am Muslim. I want them to know before they put their foot in their mouth.*
*”*
(Participant 2:12, Male, PS)

By contrast, other participants anticipated that engaging in R/S discussions could engender a positive response and possibly strengthen the therapeutic alliance.


*“I let it be patient-driven. If I notice [them wearing something religious] I will ask out of my own curiosity. It is likely important to them.”*
(Participant 3:18, Female, Psychiatrist/Fellow)

#### 3.4.3. Case-dependent Decisions Sub-Theme

Another issue that appeared to influence staff motivation to discuss R/S or make a referral was based on case-specific considerations. Examples included thinking about patients’ mental status, their current life situation, or the relationship between the patient’s experienced trauma and R/S beliefs.


*“One patient I met two times a week. Adding someone may confuse them. Later I would refer to a chaplain.*
*”*
(Participant 3:19, Male, Psychiatrist/Fellow)


*“I wouldn’t refer if the trauma is associated with church. I wouldn’t want to re-expose them to the trauma.*
*”*
(Participant 3:24, Female, Psychiatrist/Fellow)


*“If a patient’s mother is in hospice or if the spouse has recently died…if they are visibly upset, I try walking [them] over to [the chaplain]…I say to them that,*
*‘“You don’t have to believe in God but can talk to [the chaplain] anyway.*
*’”*
(Participant 1:2, Male, PS)

One participant noted referring patients to chaplaincy services when the participant felt overwhelmed by the scope of the patient’s problems or when addressing them would require more time than could be allotted to a single patient.


*“Sometimes when I’m meeting with a patient and they are telling me about the various hurts and pains, it becomes very extensive, and it feels like it’s too much for me. So I ask if they’d like to see a chaplain.*
*”*
(Participant 1:5, Female, SW)

### 3.5. Patient Motivation to Use Chaplaincy Services (Theme 3)

This theme focused on participants’ opinions about patient factors believed to promote uptake of chaplaincy services. It included two sub-themes: Patients’ Prior Experiences/Attitudes and Patients’ Life Situations. Unlike the previous theme, which primarily focused on the attitudes or experiences that could shape providers’ behaviors (i.e., discussing and referring), this theme concerned patient characteristics that participants believed could influence patients’ use of chaplaincy services.

#### 3.5.1. Case-dependent Decisions Sub-Theme

Participants believed that patients with prior negative R/S experiences hesitated to use chaplaincy services. There were other situations where participants believed that patients’ limited understanding of the scope of chaplaincy practice caused them to be less receptive to the idea of using chaplaincy services.


*“Some Vets have had a negative encounter with religion in the past and therefore shy away from such services today.*
*”*
(Participant 5:31, Male, SW)


*“[People at the VA have chaplaincy services] pigeon-holed as (1) religion, (2) services, and (3) last rites. Vets need to understand that it’s not religion.*
*”*
(Participant 5:32, Male, SW)

#### 3.5.2. Patients’ Life Situations Sub-Theme

Several participants mentioned that patients most often sought chaplaincy services during times of grief, loss, or transition. Participants believed these were predictable times of vulnerability and recommended that chaplains should hold group sessions at such times.


*“The chaplain] actually started a grief group which led to a spirituality group. Many patients have a lot of loss.”*
(Participant 4:27, Male, Nurse)


*“Holidays are ’particularly bad’ for many patients, so maybe doing a group around that time of year would be helpful.*
*”*
(Participant 4:28, Female, Nurse)

### 3.6. Chaplain Accessibility (Theme 4) 

When chaplains were seen as accessible, visible, and approachable, both staff and patients were often receptive to referring or initiating care with chaplaincy services. This final theme focused on the availability of chaplains to both staff and patients and influenced both the provider and patient pathways. It included two sub-themes. Chaplain Characteristics focused on how chaplains’ personal characteristics influenced the use of chaplaincy services. Chaplain Visibility concerned the nature and extent to which chaplains’ physical presence within the clinic and during clinical meetings could influence the use of chaplaincy services.

#### 3.6.1. Chaplain Characteristics Sub-Theme

Chaplains’ personalities often influenced participants’ decisions to refer patients or for some patients to self-refer to chaplaincy services. Knowing the chaplain in a more personal way and getting a sense of the chaplain’s style, trustworthiness, and capabilities appeared to encourage staff and patients alike to view chaplains as vital members of the treatment team. One example of first getting to know the chaplain in a non-therapy setting involved a patient who initiated services after learning that a particular chaplain bicycled to work. 


*“It has been helpful that [the patients and staff] know [the chaplain], the whole person.*
*”*
 (Participant 2:14, Male, PS) 

#### 3.6.2. Chaplain Visibility Sub-Theme

Chaplains’ physical presence in the clinics influenced staff referrals and patients’ use of chaplaincy services. Participants noted that having chaplains on-site, having regular office hours, and attending treatment team meetings reinforced awareness of the availability of chaplaincy services. Co-locating chaplains’ offices with the other clinicians and having chaplains join staff lunches and celebrations permitted clinicians and patients alike to gain trust in and awareness of easy access to the chaplain. Others noted that when chaplain visibility was low, patients could think the chaplain visit was for a negative reason (e.g., critically ill patient).


*“Vets are more receptive to chaplaincy services when there is a chaplain physically on-site.*
*”*
(Participant 3:4, Male, Psychologist) 


*“[The chaplain is] not in my awareness. So, [the chaplain] being in the team meetings and shar[ing] information helps me think, ‘Oh yeah, this might be worth referring to a chaplain.’”*
(Participant 3:21, Male, Psychiatrist/Fellow)


*“Seeing a priest can mean ‘Am I dying?*
*’*
*for many!*
*”*
(Participant 4:28, Female, RN)

Recommendations to promote the use of chaplaincy services included having chaplains routinely conduct follow-up visits and informing patients at intake that such a visit would occur. Of note, one topic never mentioned in any of the discussions was the fact that chaplaincy services are also available to staff.


*“Inform patients at admission that they will receive a follow-up call from the chaplain and perhaps at one of those contacts the patient will connect*
*.”*
(Participant 4:9, Female, Psychologist)

## 4. Discussion

The current study is one of the first to examine mental health providers’ attitudes about and motivations to engage in R/S discussions with their patients and to refer some to chaplaincy services within VACHS Substance Use Specialty Care outpatient clinics. One of the most compelling study findings was that staff and patient R/S attitudes and prior experiences could strongly influence—negatively or positively—R/S discussion and use of chaplaincy services. Those with previous positive experiences, more favorable attitudes, or an accurate understanding of R/S and chaplaincy services appeared more amenable to having R/S discussions with patients and making referrals to chaplaincy services. These findings are consistent with previous studies [41] and the substantial variability of use of chaplaincy services among clinical staff [28,29]. 

Our findings suggest several possible strategies for expanding chaplaincy services within SUD outpatient treatment settings. Staff could benefit from knowing the chaplain at both an academic and personal level. With respect to the former and similar to another study [42], staff had misconceptions about the potential value of R/S discussions and chaplaincy services. Notably, however, staff were eager to learn about the spiritual care that chaplains provide and wanted follow-up information from chaplains for specific cases. Other studies have noted that misconceptions, negative attitudes, and limited use of chaplaincy services could be improved through opportunities to shadow chaplains [43,44] and education about chaplains’ training and capabilities [45,46]. Establishing formal communication channels would permit chaplains and providers to inform each other about patients’ treatment plans and progress. Information or Q&A sessions could also be offered to patients to increase their understanding of chaplaincy services.

In addition to learning about chaplains’ training and approach to care, the data strongly suggest that staff and patients need to know the chaplains at a personal level. In this way, relationships and trust are built. The theme of chaplain accessibility offered compelling examples of the importance of chaplain visibility. For staff, sharing space and participating in joint team meetings can provide concrete evidence of chaplain capabilities and opportunities for chaplains to become known and trusted members of treatment teams. For patients, regular office hours and an “open-door policy” can normalize the chaplain’s presence. Chaplains would be seen as members of the substance use outpatient treatment team rather than appearing only during crises or for specific religious rituals. Group sessions with the chaplain could further demonstrate to patients that chaplains are approachable, capable, and a potentially valuable treatment resource. It appears that patients desire R/S discussions [12,13,47] and when patients and providers have regular and positive interactions with chaplains, patients’ perceived well-being and treatment satisfaction increase [48]. 

We note several study limitations. First, although virtually all the staff from the SUD outpatient treatment teams at the VACHS participated in this study and thematic saturation was achieved, the findings may only apply to this site; other themes may be identified at other VA facilities, in other regions of the US and abroad, or in other outpatient services. Second, the study provided only indirect evidence of SUD patients’ attitudes about R/S discussions and chaplaincy services. Patients should be interviewed to confirm the accuracy (i.e., credibility) of the current findings. Third, reliance upon focus group summaries rather than verbatim transcripts from audio-recorded sessions introduced the potential for misinterpreting participant comments. However, we sought to reduce this risk by carefully querying comments during the sessions and attempting to capture as many direct quotes in our notes as possible. We also sought to mitigate the potential for social desirability bias by having team members with limited or no previous professional relationships with study participants conduct the focus groups. Finally, while participants recommended several potential intervention strategies to promote R/S discussions and referrals to chaplaincy services, larger-scale studies should be conducted to confirm the findings before developing interventions to address these needs.

## 5. Conclusions

The study findings suggest that providers in outpatient SUD treatment services in the VACHS have varied attitudes and misconceptions about R/S discussions and chaplaincy services but are receptive to learning more on this subject and the notion of expanding chaplaincy services. Trust and confidence in the benefits of chaplaincy services may be improved among clinicians and patients by increasing chaplains’ accessibility and visibility within these outpatient settings. 

## Figures and Tables

**Figure 1 ijerph-19-09441-f001:**
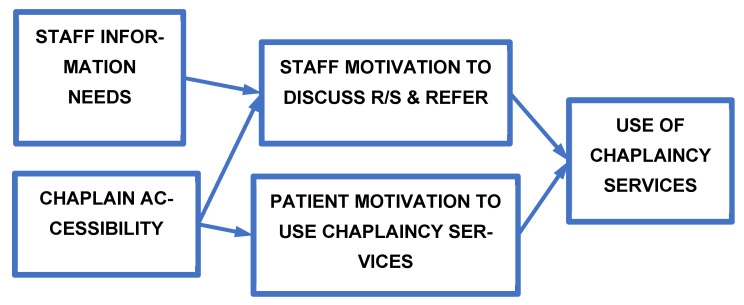
Conceptual framework of staff and patient motivation to engage in religious/spiritual discussions and Use Chaplaincy Services. The figure identifies the four themes identified and their relationship to the outcome of using chaplaincy services by veteran patients and/or service providers in the Substance Use Specialty Care outpatient clinics of the VA Connecticut Healthcare System.

**Table 1 ijerph-19-09441-t001:** Study sample characteristics (*n* = 38).

	*n* (%)
Sex	
Male	21 (55.3)
Female	17 (44.7)
Position	
Addiction Therapist	3 (7.9)
Nurse	5 (13.2)
Peer Support Specialist	16 (42.1)
Pharmacist	1 (2.6)
Psychiatrist/Fellow	8 (21.1)
Psychologist	1 (2.6)
Social Worker	4 (10.5)

## Data Availability

The data presented in this study may be made available upon request to the corresponding author.

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
