# Peer review of "Healthcare Providers’ Perceptions about the Role of Spiritual Care and Chaplaincy Services in Substance Use Outpatient Treatment"

_ijerph, 2022, doi:10.3390/ijerph19159441_

Round 1

Reviewer 1 Report

First, congratulations on the study: an innovative topic worthy of systematic study. Spirituality is a fundamental domain and is scarcely investigated.

Table 1 describes the characteristics of the respondents. It is reported "had been in their current positions for over one year. No group differences in the-matic content were noted by either participants' gender or job title" but it would be interesting to have the average figure of the respondents' clinical experience (if possible). In addition, the data show that 16 Peer Support Specialists were interviewed, which turns out to be almost 50% of the sample. Better describe why this was chosen.

The division into Themes and Subthemes is very clear. I wonder if it would be possible to better clarify before line 166 this mode of your writing. "Example: No. 4 themes will now be described subsequently divided into other sub-themes to describe qualitative results."

I wonder if it would be possible to create a table summarizing the main results/key findings for each identified theme. A sort of take home message. I am not an expert in qualitative data analysis but sometimes having a summary table clarifies the message.

No clear directions are given in the conclusion regarding how to

- Know more about the services chaplains perform (training? Conferences? Etc)

- Know chaplains more personally (how do the authors plan to limit this issue?).

I am not a native English speaker and therefore unable to give a review of the English language (but I note that the authors are all native American speakers)

Reviewer 2 Report

A minor review was requested. The suggestions are made in the attached PDF file. I have considered that these reported experiences will be used as a reference for improving the quality and access of patients to chaplain services. Also, it is required to review the word spacing and hyphenated separation throughout the manuscript.

Please, see 'https://pubmed.ncbi.nlm.nih.gov/26901281/' J Health Care Chaplain. . 2016;22(2):67-84. doi: 10.1080/08854726.2015.1133185. 

Author Response

Please note that the specific responses to Review 2 are contained within the pdf file that I received but am unable to upload as our response.

Reviewer 2

  1. “review the word spacing and hyphenated separation throughout the manuscript.”
    1. We have reviewed the formatting problems noted in the pdf version of the manuscript. Unfortunately, the extra hyphenations and deleted formats to quotations were not in the original Word or pdf files submitted. We thank the reviewer for bringing this to our attention, the issue of spacing and hyphenation is beyond our control as the authors, but we will be sure to carefully check the galley proofs.

  1. The remaining issues raised are responded to within the attached pdf file.
    1. Our responses are contained in the attached pdf file.

Reviewer 3 Report

It is a well-written article that qualitatively analyzed the medical providers’ perception of spiritual care and chaplaincy service in patients with substance use disorder through focus group interviews.

In order to improve this article, the following concerns will be required to address.

Introduction

line 61-62

It seems that the justification for the selection of research subjects is not sufficiently described.

I suggest an additional description of the reason why the participants of this study were outpatient-based faculties and the benefits of the spirituality care and chaplaincy service on substance use disorder in previous studies are needed

Materials and Methods

line 99-100, 419-420

If possible, please provide the review exemption number from IRB.

line 92

Did all occupations( psychiatrists, psychiatry fellows, psychologists, social workers, addiction therapists, pharmacists, peer support specialists) participate in 5 focus group sessions? Please describe whether there were sessions that included only certain occupations.

line 115-116

In a qualitative study, member check is used to improve the accuracy, credibility, validity, and transferability of a study. However, member check was not performed in this study. Pleases describe the additional efforts of researchers to improve the accuracy, credibility, and validity.

line 119-120

Among the participants in this study, 21 men and 17 women. The expression 'predominently male' is not appropriate.

line 144-153

It seems to be a lack of explanation for the relationship and flow with the four identified themes that contribute to use of chaplaincy.

Please describe in detail the process of the establishing the conceptual framework of the identified themes.

minor typo correction

line 48 moreso more so

There is a lot of inappropriate use of hyphens in the manuscript. Please correct.

Author Response

Reviewer 3

  1. Lines 61-62: It seems that the justification for the selection of research subjects is not sufficiently described. I suggest an additional description of the reason why the participants of this study were outpatient-based faculties and the benefits of the spirituality care and chaplaincy service on substance use disorder in previous studies are needed.
    1. We thank the reviewer for the request for further clarification. The Introduction makes the following points: (1) Many patients desire to discuss R/S needs as part of their health care. (2) Chaplains have specific expertise in addressing R/S issues, chaplaincy services have been demonstrated to have positive effects, but most existing R/S studies are confined to in-patient or out-patient palliative care/oncology settings. (3) The published research in out-patient SUD clinics is confined primarily to 12-step groups and rarely if ever led by chaplains.

We have added text to the final paragraph of the Introduction to note the rationale for conducting this study and reason for targeting the different types of healthcare providers present in out-patient SUD treatment clinics. The revised paragraph is as follows:

Despite the presence of a chaplain in the outpatient SUD treatment clinics, chaplaincy services appeared to be underutilized. Furthermore, given the limited published information about chaplaincy service utilization in outpatient settings, the study sought to understand some reasons that may account for this underutilization. Recognizing the value of R/S in mental health and embedding chaplains in the outpatient SUD service clinics [34] and reasoning that outpatient mental health providers play a potentially critical role in linking patients to chaplaincy services, we conducted focus group discussions with these providers to understand their perceptions about R/S and chaplaincy service utilization in outpatient SUD treatment clinics and the potential barriers and facilitators to use of these services.

  1. Lines: 99-100, 419-420: request for exemption number
    1. We received a Letter of Determination from the Associate Chief of Staff for Research (overseeing the IRB) stating that this project was done for quality improvement and is not research.  It therefore did not require review by the IRB or R&D committee.  Publication or presentation describing the study is permitted. We have modified the text accordingly:

The study was deemed a quality improvement study and therefore exempt from human subjects research review by the IRB at VACHS although publication or presentation describing the study is permitted.

  1. Line 92: Did all occupations (psychiatrists, psychiatry fellows, psychologists, social workers, addiction therapists, pharmacists, peer support specialists) participate in 5 focus group sessions? Please describe whether there were sessions that included only certain occupations.
    1. We have modified the first paragraph of 2.2 to further describe focus group composition. We noted that, with the exception of one group, the remainder were heterogeneous. We provided a rationale for our decision to conduct one session with only PSs.

  1. In a qualitative study, member check is used to improve the accuracy, credibility, validity, and transferability of a study. However, member check was not performed in this study. Pleases describe the additional efforts of researchers to improve the accuracy, credibility, and validity.
    1. We agree that it is ideal to include member-checking as an essential component of any qualitative research project. However, as is often the case, this step is not performed (e.g., participants are no longer available, time or geographic constraints make it logistically impossible). In fact, many of the staff had either changed jobs within the VA system or had completed their training fellowship by the time the data were analyzed. Hence, it was impossible to member-check.

  1. Among the participants in this study, 21 men and 17 women. The expression 'predominently male' is not appropriate.
    1. We have changed the text to:

The majority of the sample was male (Table 1).

  1. It seems to be a lack of explanation for the relationship and flow with the four identified themes that contribute to use of chaplaincy. Please describe in detail the process of the establishing the conceptual framework of the identified themes.
    1. The conceptual framework was developed as part of the thematic analysis and served to account most parsimoniously for all the content of the data set as well as the relationships between themes. The revised text is:

As part of the thematic analysis, we identified developed a conceptual framework that was firmly grounded in the data. We identified two pathways, four themes, and the relationship between themes that could lead to use of chaplaincy services. The provider-initiated pathway involved staff motivation to engage in R/S discussions with patients and/or to refer them to chaplaincy services; the patient-initiated pathway pertained to participants’ perceptions about patients’ personal motivation to use chaplaincy services and independent of the patients ever having had an R/S discussion with a provider. These two pathways shaped use of chaplaincy services and subsumed four themes: (1) Staff Information Needs which, in turn, influenced (2) Staff Motivation to Discuss and Refer, (3) Patient Motivation to Use Chaplaincy Services which directly influenced whether patients would seek such services, and (4) Chaplain Accessibility which directly influenced both staff and patient motivation and indirectly influenced the outcome of using chaplaincy services (Figure 1)… These themes will be described and subsequently divided into sub-themes below as part of reporting the findings from the thematic analysis.